# Antimicrobials as cornerstones and quick fixes Zimbabwean in healthcare and society: Health practitioners´ critical reflections on two stories of antimicrobial use as part of antimicrobial resistance (AMR) education

**Martin Mickelsson**[1,2]*, **Tungamirirai Simbini**[3]

**1** Department of Earth Sciences, University of Gothenburg, Gothenburg, Sweden, **2** Department of Women´s and Children´s Health, Uppsala University, Uppsala, Sweden, **3** Department of Biomedical Informatics and Biomedical Engineering, University of Zimbabwe, Harare, Zimbabwe

* martin.mickelsson@gu.se

## Abstract

Antimicrobials are often presented as key for the sustainability of healthcare as these pharmaceuticals are viewed as critical resources for much of modern medicine. Communicable diseases are a major contributing factor to morbidity and mortality in developing countries. The emergence of antimicrobial resistance (AMR) thus poses a significant challenge to global public health towards controlling these diseases and the SDG 3 Good health and well-being promoting calls for shared responsibility in preserving antimicrobials. This paper aims to explore health practitioners' understandings of the role of antimicrobials in healthcare and society and how this could inform antimicrobial resistance (AMR) education. Using a qualitative participatory research methodology, two participatory research workshops formed the empirical basis for the study and included 25 health practitioners from two major Zimbabwean central hospitals in the latter half of 2023. The focus of the workshops was on participants´ engagement with and discussions of two conceptual stories of antimicrobials in healthcare and society, as *cornerstones* which are key to the sustainability of healthcare and viability of modern medicine, and *quick fixes* that are used to mitigate but not resolve deeper and structural challenges as part of the Zimbabwean healthcare and society. During the workshops research data was collected through audio recordings supported in the analysis by contemporary field notes as well as written documentation created by the workshop participants. Three interconnected themes were identified as part of the results outlining how participants operationalised the two stories as part of AMR education. These included (i) preventing common infections, (ii) addressing risk factors, and (iii) engaging with societal inequalities. A key result was how the participating health practitioners highlighted the need to reduce reliance on antimicrobials which in turn necessitates a shift in focus towards preventive health

**Data availability statement:** Anonymised data that supports the results of the paper are attached as supplementary data.

**Funding:** This work was supported by the Swedish Research Council (Grant 2020-04567 to MM). The funders had no role in study design, data collection and analysis, decision to publish, or preparation of the manuscript.

**Competing interests:** The authors have declared that no competing interests exist.

actions such as improved hygiene, better water and sanitation as well as improved infection control. Such preventive efforts were furthermore linked in the participants´ discussions to structural challenges, including poor housing, limited access to clean water and inaccessible health care that was quoted as crucial to reduce infection risk and thus mitigate the need for antimicrobials in the first place. Bringing the identified themes and in-depth participant discussions together in the discussion, the paper presents a mirror model of antimicrobials in healthcare, highlighting how they are essential resources and cornerstones for healthcare while simultaneously and perpetuating systemic challenges in healthcare and society. The integration of this co-created knowledge as part of AMR education would contribute to a shift from the prevalent focus on preventing resistance to also consider the prevention of infections and the need for antimicrobials, including understanding and addressing the root causes of infections. Such a holistic approach to AMR education could promote more sustainable health practices, linking AMR challenges with broader societal and systemic challenges as part of more effective health educational efforts.

## Introduction

Antimicrobial resistance (AMR) is a major global public health crisis with developing countries in major crises due to the high burden of infectious disease. In 2021, out of 7.75 million deaths attributed to bacterial infections, 4.71 million suffered from AMR associated conditions such as lower respiratory tract, blood stream or intra-abdominal infections [1]. This paper aims to explore Zimbabwean health practitioners' understandings of the role of antimicrobials in healthcare and society and how this could inform antimicrobial resistance (AMR) education. AMR is defined as the ability of microorganisms to resist the effects of pharmaceuticals once effective at killing them or inhibiting their growth [2–6]. Resistance results from evolutionary adaptation where microbes develop mechanisms that counter the effects of antimicrobials with the consequence that existing medical treatments become less or even ineffective [7,8]. While AMR occurs naturally the overuse and misuse of antimicrobials as part of especially healthcare and agricultural practices means that this adaptation process has been accelerated [3,4,6]. Meanwhile, Caudell [9] notes that factors driving AMR in context are seldom thoroughly studied and will vary substantially between the part of the world and within regions and countries. These factors encompass social determinants of health, limited access to healthcare, poor water and sanitation as well as food insecurity, which are key in shaping antimicrobial (mis)use and the development of resistance [10,11]. While individual misuse is important in addressing AMR, social determinants of health highlight how addressing social inequalities that increase infections and need for antimicrobials is equally important. This difference of approach is also reflective of how AMR education often focuses on the emergence of de-novo mutations and acquisitions of resistance genes, i.e., the first half of the resistance process [12]. The spread of antimicrobial resistance as a further dimension

of AMR, involves the distribution of resistant strains through different environmental vectors such as humans, animals, agriculture and flows of water [9].

Further conceptual depth can be afforded by considering risk factors [13–15]. Individual and behavioural antimicrobial misuse represents a direct risk for AMR while it can be argued that such behaviours do not suddenly appear but are rather shaped by structural conditions and social determinants of health, constituting indirect AMR risk. Effective AMR education would thus necessitate holistic strategies that engage with both the direct risk of individual behaviours and the indirect risks of social determinants of health. Communicable diseases are a major cause of morbidity and mortality in Zimbabwe. HIV/AIDS, Tuberculosis and Malaria being the major contributors. The emergence of AMR in the control of communicable diseases poses a significant challenge to primary health care in the country [16]. Zimbabwe's health care delivery is built on a primary health care approach aimed at achieving Universal Health Coverage (UHC), with the public sector serving as the primary provider. The country has 1,869 primary health facilities, staffed by Primary Care Nurses (PCNs) and State Registered Nurses (RGNs), who serve as the first point of contact for communities. There are 55 district hospitals where doctors are initially deployed, and these hospitals refer patients requiring specialized care to eight provincial hospitals. At the apex of the referral system are five central/teaching hospitals, which operate at the quaternary level, providing advanced specialist services [17]. Like many other countries, Zimbabwe continues to face significant challenges related to antimicrobial resistance (AMR), including multidrug-resistant tuberculosis (MDR-TB), anti-malarial resistance, and *Neisseria gonorrhoeae* drug resistance. Key contributing factors include the widespread use of empirical treatment for suspected bacterial infections, limited diagnostic capacity, increased use of substandard antimicrobials, poor adherence to treatment protocols, and misinformation [18].

Furthermore, health-seeking behavior and adherence to treatment in Zimbabwe are impacted by cultural beliefs, traditions, and societal norms, creating barriers to effective antimicrobial use. Decisions of when to seek healthcare and how to use prescribed medicines can be as much a decision formed by family, community, culture and religion as an straightforward individual with some communities in Zimbabwe discourage seeking of conventional healthcare while instead relying on traditional healing methods [19,20]. Additionally, taboos connected to gender dynamics play a role in health-related decision-making and limiting the ability of women to seek health care with impacts on maternal care and the health of women and children, including higher risk for infections [21,22].

This paper presents the results from workshops conducted with Zimbabwean health practitioners at Harare's two major teaching hospitals, Sally Mugabe and Parirenyatwa Central Hospitals. These institutions play a critical role in training the majority of the country's healthcare workforce. The study was carried out in the latter half of 2023, during which volunteer health practitioners engaged in discussions and reflected on two narratives of antimicrobial use—one portraying antimicrobials as essential cornerstones of treatment and the other as quick fixes for health challenges. An argument commonly forwarded is how a health care system such as Zimbabwe's will be sustainable without effective antimicrobials, drawing on ideas of these pharmaceuticals as the cornerstones of modern medicine [2,3,5,23,24]. This argument draws on the idea that diminishing antimicrobial effectiveness means increased antimicrobial resistance and that global development, especially human and animal health, is under threat [25–27]. If not addressed, AMR is posed as the end of the antimicrobial era and the emergence of a post-antimicrobial era of incurable infections and unsafe surgeries [28–30]. Furthermore, in these accounts, antimicrobials become a global public good and condition for sustainable development, especially SDG 3: Good health and well-being [3,4]. Without the essential resource of antimicrobials, the developmental achievements of the last 15 years are said to be in jeopardy with calls for people around the globe to take up a shared responsibility to preserve these antimicrobials [2,5,31].

Coupled with the effort to preserve antimicrobials is an emphasis on the development of antimicrobials through business models and stimulation of research and development by decoupling return on investment from sales and price to produce what is presented as new and necessary technologies [2,4,32,33]. Exploring the role of antimicrobials in healthcare and society, Denyer-Willis & Chandler [34], and Dixon et al. [35] argue that antimicrobials are more than an

innocuous technical addition. Their presence enables specific approaches to health while limiting others, facilitating a shift towards therapeutic healthcare, more effectively treating infections, lessening the need for preventive measures and allowing for time to recuperate after an infection. As such, the use of antimicrobials is seen as being part of the emergence and perpetuation of societal structural challenges that over the long term undermine the provision of care, equal access to sanitation and safe water. Instead of addressing these challenges, pharmaceuticals are provided once infection has occurred. A result of how antimicrobials are used is how to correct hygiene and health issues caused by entrenched inequality in society. Denyer-Willis and Chandler [34] emphasise how antimicrobials have become a pharmaceutical replacement for adequate healthcare and societal health-promoting services. Moreover, Beisel et al. [36], Rogers Van Katwyk et al. [37] and Hays et al. [38] have illustrated how reactions to AMR often involve a doubling down on antimicrobials as technological solutions to circumvent social, sanitation, and water quality problems driving vulnerability to resistant pathogens.

This paper draws on these two different but interrelated stories of antimicrobials in healthcare and society as both cornerstones and quick fixes. Contextualising the stories as part of discussions on social determinants of health and risk factors, the paper emphasises how medical, economic, and social conditions impact antimicrobial use and by extension AMR. Including notions of antimicrobials as quick fixes enrich discussions beyond binaries of rational/irrational use and individual responsibility to account for both individual and structural drivers of antimicrobial use and dependence. This forms the basis for AMR education frameworks that address both immediate and structural drivers of antimicrobial dependence. By engaging health practitioners in discussions on these two stories current AMR challenges become opportunities to reflect on the focus in AMR education on the use of antimicrobials and give equal attention to the societal challenges that drive the need for antimicrobials through increased rates of primarily common infections [39,40].

Based on the aim of exploring Zimbabwean health practitioners' understandings of the role of antimicrobials in healthcare and society and how this could inform antimicrobial resistance (AMR) education through two workshops held at major hospitals during the latter part of 2023, two research questions are formulated.

- How do Zimbabwean health practitioners at two major hospitals in Harare operationalise different understandings of the role of antimicrobials in healthcare and society?

- How can these operationalised understandings inform AMR education in Zimbabwe?

- In what ways can integrating health practitioners' reflections on societal challenges that drive antimicrobial use contribute to more effective antimicrobial resistance (AMR) education strategies?

## Materials and methods

Using a qualitative participatory research methodology, two participatory research workshops (PRW) with Zimbabwean health practitioners formed the basis for the study [41–43]. PRW focuses on the active involvement of participants in knowledge co-creation, drawing on their expertise and experiences [43]. This study was conducted at two major teaching hospitals in the country, providing an optimal setting to examine the values and principles of antimicrobial resistance (AMR) in both clinical practice and the training of future health practitioners. A voluntary sampling approach was employed, where participants were provided with the project outline and recruited based on their willingness to participate. The study aimed to recruit 13 participants from each of the teaching hospitals, a total of 26 from the two teaching hospitals. Twenty five (25),13 Parirentyatwa and 12 at Harare (Sally Mugabe) central hospital.

Participants and researchers were, through the workshops contributing to knowledge co-creation [42]. In this study, the method is used to engage health practitioners in reflective discussions that explore diverse understandings and approaches to antimicrobial use and resistance. Having participants drawing on experiential knowledge provides insights into how they operationalise their understandings of antimicrobial use and resistance and relevant social factors [41]. The

methodological approach is relevant to study as it creates a space for health practitioners to engage in critical reflection on experiences of antimicrobial and the societal factors shaping their use. Furthermore, as the challenges of AMR differ across regions and contexts, the method enables the joint investigation, with participants, of AMR challenges in Zimbabwean healthcare and society. While generating empirical data the workshops also develop participants´ knowledge and capacities to critically engage with AMR education, contributing to bridging the gap between knowledge and practice.

As a research method PRW have been previously utilised in research in Southern Africa [44,45] providing for the method's contextual relevance. The studies explored how the complexities of sustainability challenges could be addressed through PRW, facilitating knowledge co-creation and adaptation. Results illustrated how engaging participants around their lived experiences enabled them to better contextualise sustainability challenges and develop more contextually adaptive approaches to addressing sustainability challenges. The two stories of antimicrobials utilised in the workshops were developed based on previous research detailed in the introduction and drawing on contrasting but interconnected research perspectives on how antimicrobials are used, valued and ultimately relied on as part of especially health care [2,3,5,23,24,34,35]. With the aim to create a more comprehensive understanding on the role of antimicrobials in healthcare and society, the two stories were prior to the workshops presented at research seminars and conferences and further developed based on peer feedback. Discussions around Fig 1 were initiated by first having participants reflect together on the conditions of AMR education based on their experiences as health practitioners. The two stories were then presented, and participants were asked to discuss how the stories related to their experiences and understandings of antimicrobials and AMR. As such, the approach aimed to facilitate engaging with the stories and contextualise them in the Zimbabwean health care context. Development and preparation of workshops drew on previous research [44–46] and was guided by the research aim. Empirical materials were generated during three-hour workshops at Parirenyatwa University Hospital and Sally Mugabe (Harare) Hospital, both located in Harare, Zimbabwe. Due to significant health challenges related to AMR and southern Africa being identified as one of the regions most at risk from AMR, makes Zimbabwe a highly relevant study site for this research. The prevalence of comorbid conditions in the country, including HIV/AIDS and tuberculosis (TB) exacerbating AMR risks also makes Zimbabwe relevant for studies on AMR education. Workshops were co-facilitated by the authors. The hospitals were selected as they are central to both the Zimbabwean health care and national efforts to

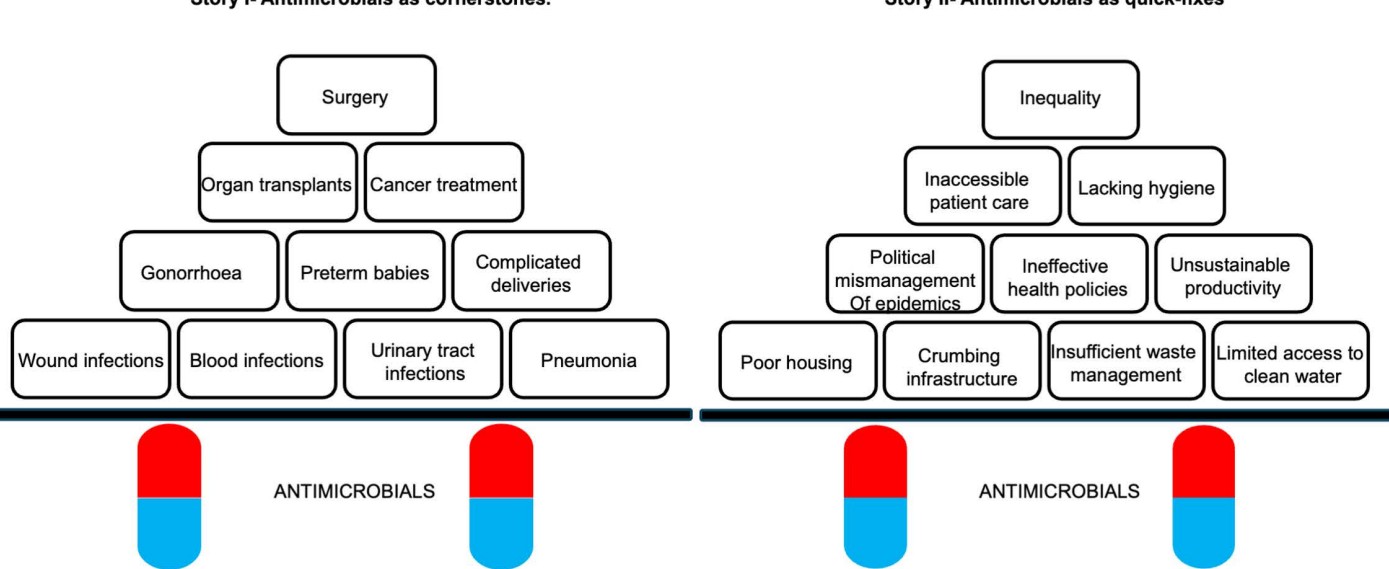

**Fig 1. Two stories of antimicrobials in healthcare and society.**

address AMR. Together the hospitals provide care for a broad spectrum of Zimbabweans both in Harare and throughout the country as many more severe cases are referred to these hospitals. Qualitative data was generated through audio recordings supported in the analysis by contemporary field notes by the authors as well as written documentation created by the workshop participants and collected at the end of the workshops.

The study included 25 health practitioners conveniently sampled from each hospital, 10 medical doctors (4 women, 6 men), 2 nurses (2 women) and 1 pharmacist (1 woman) at Parirenyatwa hospital: 8 medical doctors (3 women, 5 men), 2 nurses (2 women) and 2 pharmacists (2 women) at Sally Mugabe (Harare) hospital. Workshops were held during the latter half of 2023. Participant inclusion criteria included health practitioners (medical doctors, nurses and pharmacists) with an interest in AMR or are working with AMR issues in practice. The inclusion criteria of participating health practitioners were based on their practices in relation to health and especially professional engagement with AMR and AMR education. Diversity was sought regarding gender, age, and disciplinary background with health practitioners from paediatrics, medicine, surgery and pharmacy. The inclusion criteria included pharmacists to strengthen drug prescription and dispensing practices impacting on access and excessive use of antimicrobials. Participant exclusion criteria included non-practising health practitioners, practitioners who stated prior to participating that they could not participate in a majority of the workshops, practitioners who did not want to participate in the study, and practitioners who were ill or had at least 5 years without practicing medicine. Recruitment of participants at respective hospital started on 30 June 2023 and ended on 27 July 2023. The workshops generated spaces for engagement and knowledge co-creation among health practitioners across medicine, nursing and pharmacy [47–50]. The participatory research workshop method provided structure for the data collection and created conditions for knowledge co-creation. Workshops were audio recorded and transcribed by the authors with support from contemporary notes for improved depth and accuracy in the transcripts. The data analysis was conducted by the authors using an inductive content analysis method [51] which offers theoretical flexibility in analysing qualitative data. The analysis moved through several steps: firstly, reading through the transcriptions to create an overview of the data followed by an in-depth re-reading. Secondly, structuring the data by identifying key sentences relevant to the research aim and questions. Thirdly, creating codes from key sentences, as crucial meaningful words or phrases as basis for analytical categories and sub-categories. Consideration was given to emphasise and make clear the participant voices on the research topic, carried forward into the presentation of research results, supporting results with quotes.

The study instrument was validated by pilot participatory research workshops at separate universities namely Midlands State University, Zimbabwe and the Department of Nutrition, Dietetics and Food Sciences at University of Zimbabwe. The validation was focusing on linguistic accessibility and comprehension and drawing on peer-feedback to refine clarity and effectiveness. Validation was integrated into designing and preparing the workshops to align them with the aims of the research and draw on previous use of workshops in the region. Furthermore, data generation methods and generated data were validated by structuring these instruments to effectively account for participants' reflections and discussions as part of the workshops.

### Ethical considerations

Ethical approval for the study was received from the Medical Research Council of Zimbabwe (MRCZ/A/2920). Furthermore, as part of designing and planning the study and workshops with health practitioners the authors sought and obtained permission from the clinical directors at respective hospital; Parirenyatwa hospital and Sally Mugabe (Harare) hospital. Throughout the study, authors adhered to principles of informed consent and voluntary participation with participants provided consent forms and information sheets as part of the workshops. The information sheet presented participants with the aims of the research, and the workshop process. Furthermore, the information sheet detailed how participant confidentiality would be maintained, what research data would be collected and how the data would be managed. Participants were provided opportunities at the beginning of the workshops to raise possible questions and

concerns either regarding the research or the workshop process. This was followed by a thorough reading of the consent form and signing by the participants. During the workshops further opportunities were offered to raise and address any subsequent questions that might emerge on the part of the participants. To maintain confidentiality participants were coded with numbers, and no participant names were recorded as part of data collection. Throughout the study, the global code of conduct for research in resource-poor settings was adhered to [52].

## Results

In this section, we present results from the study, outlining workshop discussions on the two stories of antimicrobials and antimicrobial use, Fig 1 Throughout the workshop discussions, participating health practitioners reflected on and highlighted how the two stories were not oppositional but rather interconnected, together providing a more nuanced and deeper understanding of the role of antimicrobials in healthcare and society.

Three interconnected themes were identified in the participants´ discussions highlighting how Fig 1 could be operationalised in AMR education as part of *preventing common infections, addressing risk factors* and *engaging with inequalities.*

### Preventing common infections

The first theme has participants highlighting how Fig 1 can be operationalised as part of AMR focusing on preventing common infections such as wound infections, blood infections, urinary tract infections and pneumonia. As seen in the following quote these infections represent the most common uses of antimicrobials and thus are the areas where the largest dependency on antimicrobials lies within the Zimbabwean healthcare and society.

*The highest number of people who need antibiotics they've got wound infections, urinary tract infections and pneumonia. Then we go up gonorrhoea, then we go up organ transplants then to surgery.*

*Participant 3, Group 1, Harare workshop*

Without antimicrobials, these common infections will be significantly more difficult to treat once they emerge. To these considerations, participants added gastrointestinal infections which they describe as common areas of antimicrobial use as well as respiratory tract infections as an important cause of pneumonia which requires antimicrobials as seen in the following quote,

*There are also other conditions that may complicate it like respiratory tract infections because it can lead to pneumonia as it is a result of the complication of the upper respiratory tract infection.*

*Participant 4. Group 1, Parirenyatwa Workshop*

Furthermore, participants highlight in their discussions how the dependency on antimicrobials in cancer treatment, organ transplants and surgeries is based on the risk for common infections such as wound and blood infections, resulting from these medical practices. As seen in the following quote, care in these areas is often linked to the use of antibiotics with the purpose of preventing infections while such reliance highlights how awareness for AMR and when and how to prevent infections becomes key.

*We need to be aware of antimicrobial resistance in everything because it plays a role. Be it in surgery, when you are given antibiotics, pre-operation, post-operation, organ transplant, in cancer treatment, but in everything else, when you look at the wound infections, yes, you are going to do dressings, and give antibiotics.*

*Participant 2. Group 2, Parirenyatwa Workshop*

As such, as we consider cancer treatment, organ transplant and surgery, the concern and focus when operationalising Fig 1 in AMR education should, according to the participants, stay with the key message of the importance of common infections in terms of generating the need for and dependency on antimicrobials.

Throughout these discussions, participants note that operationalised in AMR education, Fig 1 can be used to raise awareness regarding the importance of preventing infections in the first place using widely accessible methods such as hand hygiene and waste disposal, thus reducing the need for antimicrobials as seen in the following quote.

*So there are ways to prevent infection in the first place, hand hygiene, and also proper disposal of waste. So it's important that the patient understands the two sides of the story.*

*Participant 2, Group 1, Harare Workshop*

These preventive efforts to avoid various infections are crucially available to patients themselves including hand hygiene, clean bathrooms and the proper disposal of waste as well as making sure to take care of and re-apply wound dressings in a correct way. In addition, as seen in the following quote, the participants are viewing the operationalising of the stories as a way to shift AMR education from preventing resistance to preventing infection as it is the primary source of need for antimicrobials and also the area where reduction of infection can be most easily and effectively achieved.

*We were initially first focused on how do we prevent resistance? How do we prevent resistance? But later on, we moved on to what if we actually preventing infection in the first place? That's more important. And I think that if you start off on that step by step, if you start off right at the basics, you can actually achieve true, true change rather than just we learned about a few things. But if you really get down to the nitty gritties and the basics of it, you can actually achieve bigger, bigger change. So, if we prevent infection in the first place, that way we can preserve antimicrobials for the greater purpose.*

*Participant 3, Group 2, Parirenyatwa Workshop*

As noted in the quote, operationalising Fig 1 in efforts to prevent infections can have a significant change to not just the use of antimicrobials and the risk of AMR but to the general health status of communities by limiting their susceptibility to infection.

**Addressing risk factors**

In the second theme, participants emphasize the importance of preventing common infections and operationalizing Fig 1 as a key component of AMR education in addressing risk that contribute to infections. This is critical, as these factors not only increase the burden of disease but undermines prevention efforts. As illustrated by the following quote poor housing conditions, inadequate access to clean water, and insufficient waste management expose communities to infections such as tuberculosis, typhoid, and cholera, increasing the need and reliance on antibiotics.

*I think these are the risk factors. These are the risk factors. I have got poor housing, you are exposed to TB, crumbling infrastructure, poor clinics and so forth. So those are the very risk factors. That causes the infections. That causes infections and the need for antibiotics. We could fix that. Then, you know, if you have got limited access to clean water, you are exposed to typhoid and cholera. Now and again we experience cholera. No reliable tap water and so on. Insufficient waste management.*

*Participant 1, Group 1, Harare Workshop*

Addressing these risk factors is thus key to understanding how communities become more dependent on functional (health) infrastructure, accessible patient care, effective health policies and political management of epidemics. Participants exemplify how poor hygiene and sanitation in communities make it more likely to be exposed to diseases such as cholera and typhoid indicating that Fig 1 can be operationalised in AMR education to focus on the need for societal transformation beyond the healthcare sector in addressing these risk factors, especially safe and clean water as well as better waste management in an effort to prevent what causes infections and the need for antimicrobials. Participants further emphasise that to address these risk factors as part of AMR education, there is a need to view the patient beyond the clinic and the specific symptoms on display, as illustrated by the following quote.

*So, like I was saying, poor housing, crumbling infrastructure, that issue of waste management predispose to infections. If you drink the water and now you have a stomach ache, they just come and be given antibiotics. We don't even look at what's causing that. Which water are they drinking, and from where is the source of water? For us to address that, there is a need for proper history taking.*

*Participant 1, Group 2, Parirenyatwa Workshop*

To this end, Fig 1 is operationalised in AMR education to push for the need to take a history outside of the healthcare system, to understand where the patient is coming from, why they have infections and symptoms that require the use of antimicrobials and whether there are environmental conditions in place that are likely to cause further patients to come in. As seen in the quote below, participants argue that addressing these risk factors is thus key to reducing antimicrobial use.

*Once we are able to control these risk factors the use of antibiotics will be very much reduced. The chance of us getting resistance will be much less.*

*Participant 4, Group 2, Parirenyatwa Workshop*

Operationalising Fig 1 in AMR education would thus enable a greater focus on controlling the risk factors driving infection and thus mean a significant reduction of antimicrobial use and the emergence of resistance. In relation to communities, this includes an emphasis on water, sanitation and hygiene (WASH) as seen in the following quotes.

*We need to be proactive rather than reactive, so people remember that they should wash their hands. The purpose of AMR education is to minimize the need for antibiotics and then we eliminate resistance where possible.*

*Participant 5, Group 1, Harare Workshop*

*I think water and sanitation play a major role in terms of infection control. Because what we eat and how we dispose of it is actually very important.*

*Participant 3, Group 1, Parirenyatwa Workshop*

## Engaging with inequalities

As part of the third theme, participants went further in mapping the causes of the risk factors causing the infections driving much of the need for antimicrobials, operationalising Fig 1 in *engaging with inequalities* as part of AMR education. Highlighted in the following quote, participants argued that engaging with social inequalities becomes a critical issue extending both to the everyday challenges of water, sanitation, waste and poor housing as well as inaccessible healthcare and infrastructural conditions and ineffective health policies affecting many patients.

*The moment you remove inequalities, everything underneath disappears. Because limited [healthcare] access is a form of inequality. You don't talk about limited access when you are on the other side of town, because everything is there. You only talk about limited access on this side of town. So everything underneath there is a form of inequality.*

*Participant 5, Group 2, Harare workshop*

This quote highlights the need for AMR education to address systemic inequalities that shape health outcomes. By tackling these disparities, it becomes possible to create more equitable healthcare access, improve living conditions, and ultimately reduce the burden of infections that drive antimicrobial use. To this end, Fig 1 can in the view of the participants be used in integrating social inequalities as a focus and consideration for AMR education, linking the sustainable health challenge of AMR with social sustainability challenge of social inequality, which is further highlighted in the following quote,

*If that [inequality] part is solved, we have solved probably 40–50 per cent of AMR, so that the education becomes easier. So it's easier for me to say, okay, don't take antibiotics when you're not prone to any infection. But if you're fighting the infection every day in your day-to-day life. If we solve that, that's where the government comes into it and policies as well, directly towards either our housing structures, whether just sanitation removal, complete sanitation removal, and delivery of water. So if those things are done at the government level, we can then come in with education.*

*Participant 2, Group 1, Harare workshop.*

The quote emphasises the idea is that addressing the causes of social inequality, such as access to clean water, improved sanitation, and better housing can be crucial in limiting AMR. As seen in these quotes, engaging with inequalities as part of AMR education becomes an important purpose as it would lessen the pressure of common infections on patients and healthcare while also opening for further educational efforts in responsible antimicrobial use as communities are not living with ongoing infection risk on a daily basis.

## Expanding AMR education

Throughout the three themes, there is in the participants´ discussion an expansion from primarily focusing on preventing resistance to also focusing on preventing infection as it is the primary source of the need for antimicrobials and also the area where reduction of infection can be most easily and effectively achieved as seen in the following quote,

*The two sides of the story, we could integrate them into how we educate our patients. So we tell them that we're giving you this drug because it is crucial. It will help you recover. But at the same time, we also teach them regarding prevention, like what we were saying, basic hand hygiene, such that next time they don't get an infection. So there are ways to prevent infection in the first place, hand hygiene, and also proper disposal of waste.*

*Participant 4, Group 1, Parirenyatwa Workshop*

This highlights how AMR education can go beyond the Zimbabwean healthcare system to encompass questions of societal living conditions and equality as seen in the following quote.

*It [Fig 1] becomes easier to integrate into AMR education because it's like a story that we can tell to someone and they easily understand it. So on one part, these drugs are what's holding us back. And yet on the other part, they are bad in the sense that they are sort of covering things. It's just short term, but by the same token, there are problems behind that short term. So as much as it looks like we are solving the other part and them being the bricks, that's the same way that we're destroying our infrastructure. So in that sense, it reminds someone how important they are on one side, but*

*it's a quick fix to how that importance can crumble everything. So I think the story overall would be something that you can easily give to people and they can visualise better what we're discussing.*

*Participant 5, Group 2, Harare Workshop*

As illustrated here participants present the two stories of antimicrobials in healthcare and society as an integrated part of AMR education. Through acknowledging how antibiotics can be highly beneficial short-term, especially for the individual, while at the same time have potentially significant longer-term negative societal impacts, AMR education can promote more nuanced understandings regarding the intersections of pharmaceuticals, health care needs and infrastructure. Such educational approach based on the results of the paper highlights how reflection on societal challenges of infrastructure and equality are essential complement to more traditional antimicrobial stewardship.

## Discussion

The discussion draws on the results of participants' discussions about the two stories of antimicrobials in healthcare and society that highlight antimicrobials as concurrent cornerstones for the viability of "modern'' medicine and as quick fixes for structural challenges to health. Crucially, the results detail how these two stories outlined in Fig 1 can be adapted and operationalised to expand the relevance of AMR education. There are two mirrored aspects of antimicrobials in healthcare and society that can be mapped as a conceptual model for AMR education by drawing on the conceptual frameworks and the workshop participants' discussions, as outlined in Fig 2.

The upper half of the model focuses on how most of the need for antimicrobials is the result of various common infections while STIs, contagious diseases and complicated deliveries represent an important but less extensive pressure for antimicrobial use in total. Crucially, at the top of the pyramid we find surgery, organ transplants and cancer treatments which all require the use of antimicrobials primarily to the extent they result in common infections such as wound and blood infections. The upper half thus highlights how *preventing common infections* becomes a crucial aspect of AMR education as these infections drive a large part of the need for antimicrobials in healthcare and society.

The lower half of the model emphasises how the *common infections* driving antimicrobial use are to a large extent the result of *risk factors.* These include those factors that are present in communities' everyday lives such as limited access to clean water, lacking hygiene and sanitation, insufficient waste management and poor housing as well as factors communities encounter periodically, inaccessible healthcare, crumbling health infrastructure and unsanitary working conditions [9,14]. Furthermore, these risks are crucially not spread evenly throughout society but are significantly the outcome of social and economic inequalities that channel resources and support away from many communities resulting in persistent infection risk [11]. The lower half of the model shows how *addressing risk factors* and *engaging with inequalities* that perpetuate these risk factors throughout society becomes key aspects of AMR education that aim to result in holistic and sustainable change.

The operationalisation by workshop participants of the two stories outlined in Fig 1 which constitutes the basis for the mirror model of antimicrobials in healthcare and society engages with both dimensions of resistance emergence and spread as part of the AMR challenge. While emergence centres on the development of new resistances among microbes, spread focuses on how resistance and resistant microbes are distributed via several vectors throughout living environments, including water sources as well as waste and sanitation. Crucially, resistance spread is mainly facilitated by the degradation of these environments resulting in what the participants identified as risk factors for infection and cause for subsequent antimicrobial use [12]. When environments are conducive to the spread of resistant strains, people are more likely to develop more severe infections needing treatment with stronger antimicrobials [9]. AMR education thus needs to consider how inequality, inefficient health policies and subsequent inaccessible healthcare, crumbling health infrastructure and unsanitary working conditions create environments that are rife for the spread of resistant strains such as E. coli and Cholera, especially in Low- and Middle-Income Countries (LMIC).

In line with Ahmed et al. [8] and Coque [53], the results highlight that AMR as a health challenge is partly the result of collective practices that create living environments that negatively impact the health of communities. There is thus an argument, also noted by the participants, for shifting focus in AMR education towards efforts to prevent infections and develop conditions in healthcare and society that promote community health and resilience to infection, preventing the spread of resistant pathogens in hospitals, clinics, housing as well as the water and sanitation system. AMR education come to focus on reducing the reliance on antimicrobials through holistic approaches aimed at preventing infections. This means a shift in AMR educational purposes from therapeutic to preventive.

Drawing on the argument presented in previous research [34,35] of the infrastructural role of antimicrobials as quick fixes to the challenges of Zimbabwean healthcare and health governance, exacerbated by poor sanitation, water quality and housing we present the AMR educational model outlined in Fig 2 as a way to engage with health challenges of microbial infections and resistance as integrated with societal challenges of living environments, WASH, health access and inequality. We envision that this approach to AMR education could be enacted through teaching hospitals that are key

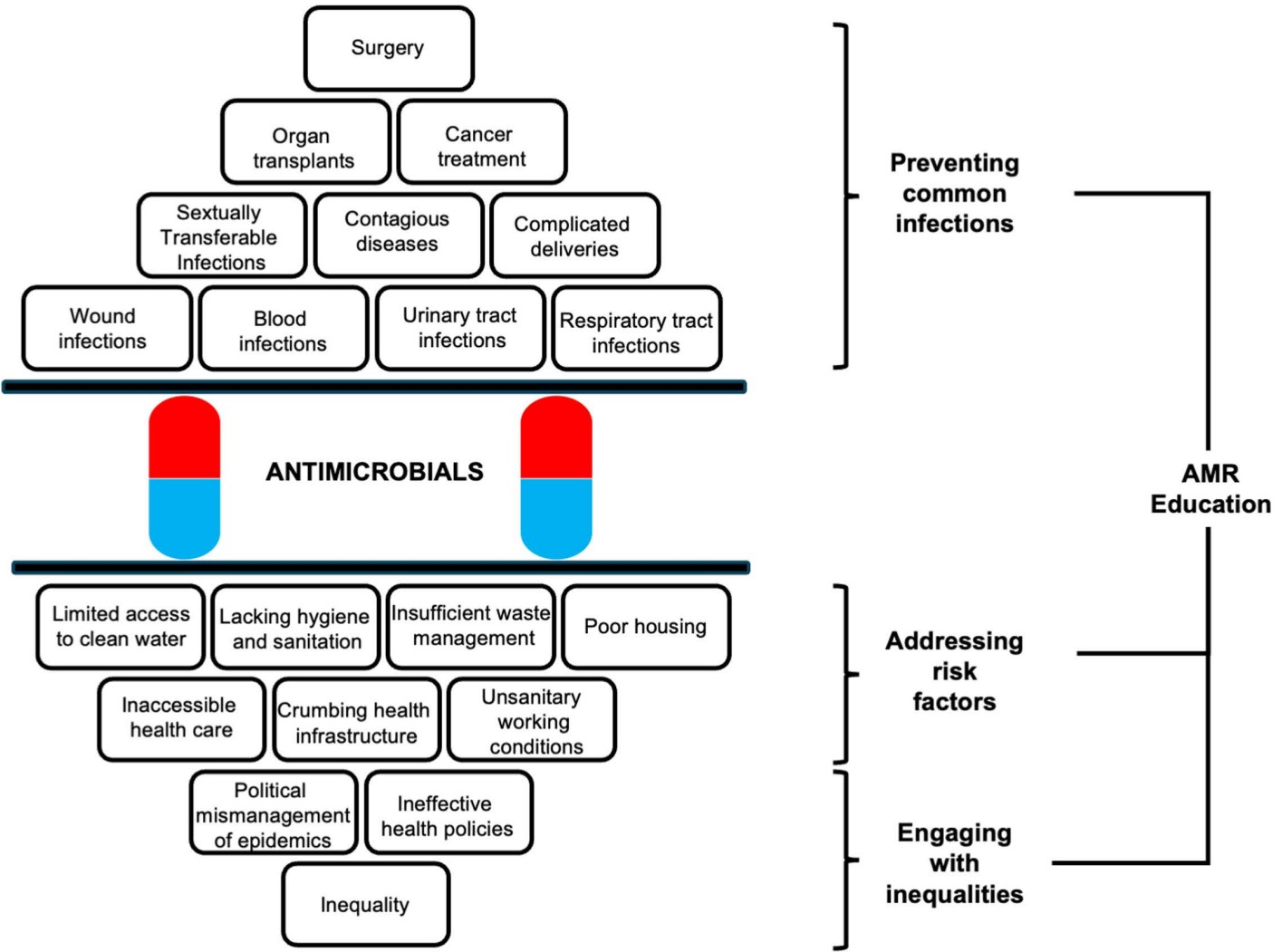

**Fig 2. Mirror model of antimicrobials in healthcare and society.**

educational settings for medical students and for the continuous professional development initiatives for practicing health workers. By integrating the AMR educational approach as part of curricula, training programs and professional development efforts teaching hospitals are envisioned as key in enacting AMR education. Furthermore, AMR education can be enacted through educational as part of patient consultations in everyday practices developing patients' awareness and understanding of AMR risks and the key role of infection prevention. The results of the papers along with their envisioned enactments can be situated in the context of national policy. With the Zimbabwean One Health National Action Plan on AMR providing a comprehensive framework for efforts in Zimbabwe to address AMR, education is positioned as a key framing component to be systematically incorporated into all public health and healthcare initiatives around AMR. While a persistent challenges to these efforts include resources, framing AMR education as significantly could address such limitations by reducing the need for antimicrobials and lengthy treatment of infections thus mitigating the spread of AMR, infections while promoting sustainable health outcomes.

## Conclusion

This paper explored health practitioners' understandings of the role of antimicrobials in healthcare and society and how this could inform antimicrobial resistance (AMR) education. Based on the results the paper details how participants come to operationalise the two stories as part of AMR education. Grounded in the results a mirror model of antimicrobials in healthcare and society is presented. The model showcases how antimicrobial use is at the same time both essential for health and an obstacle to addressing long-term health risks. We need to account for these two sides when engaging with questions of antimicrobial use and the challenge of antimicrobial resistance in AMR education. Detailing preventing common infections, addressing risk factors and engaging with inequalities as part of AMR education, illustrates how the microbial world and its consequences for human health are integrated with the social conditions under which such health challenges emerge. In these situations of intensive pressure on both healthcare and society at large, antimicrobials become an infrastructure bridging the gap between life and death when microbial infections come to pose severe health risks. As seen in the results from the participating health practitioners´ discussions these infections are the most common cases where antimicrobial treatment is needed, largely driven by poor housing, insufficient waste management, lacking sanitation and hygiene as well as limited access to clean water.

As illustrated in this paper, antimicrobial resistance includes both antimicrobials and the resistance to them, which means that knowledge practices regarding AMR can be enriched by expanding our interest and inquiry to encompass both the resistance among microbes and the antimicrobials to which the resistance is emerging. We do not claim that the mirror model represents an exhaustive characterisation of AMR education. Rather, we have attempted to encourage critical discussion and reflective thought aimed at expanding AMR education from primarily focusing on preventing resistance to also focusing on preventing infection. Based on previous research and results from the health practitioners´ workshop discussions, we show how multiple stories can be told regarding the role of antimicrobials in healthcare and society and how these can be used to expand AMR education with the aim to understand and address the sustainability challenge of resistance as interwoven with societal challenges of providing communities with access to clean water, sanitation and waste management along with good housing, accessible healthcare, sanitary work conditions and ultimately the opportunity to live in an equal society.

## Supporting information

**S1 Data. Transcript of facilitated group discussion with Zimbabwean health practitioners (medical doctors, nurses and pharmacists) as part of participatory research workshop on AMR education focusing on two stories of antimicrobials and antimicrobial use.**
(PDF)

**S1 Checklist. Inclusivity in global research.**
(DOCX)

## Acknowledgments

We acknowledge the health practitioners (medical doctors, nurses and pharmacists) at Parirenyatwa University Hospital and Harare Central Hospital, who contributed greatly to the research through their part in the participatory research workshops.

## Author contributions

**Conceptualization:** Martin Mickelsson, Tungamirirai Simbini.

**Data curation:** Martin Mickelsson, Tungamirirai Simbini.

**Formal analysis:** Martin Mickelsson.

**Funding acquisition:** Martin Mickelsson.

**Investigation:** Martin Mickelsson, Tungamirirai Simbini.

**Methodology:** Martin Mickelsson, Tungamirirai Simbini.

**Project administration:** Martin Mickelsson.

**Resources:** Martin Mickelsson.

**Software:** Martin Mickelsson.

**Validation:** Martin Mickelsson.

**Visualization:** Martin Mickelsson.

**Writing – original draft:** Martin Mickelsson.

**Writing – review & editing:** Martin Mickelsson, Tungamirirai Simbini.

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
