## [Decision Letter · Decision Letter 0]

PGPH-D-24-02902

Antimicrobials as cornerstones and quick fixes in healthcare and society: health practitioners´ critical reflections on two stories of antimicrobial use as part of antimicrobial resistance (AMR) education

Dear Dr.Martin Mickelsson,

Thank you for submitting your manuscript to PLOS Global Public Health. After careful consideration, we feel that it has merit but does not fully meet PLOS Global Public Health’s publication criteria as it currently stands. Therefore, we invite you to submit a revised version of the manuscript that addresses the points raised during the review process.

We look forward to receiving your revised manuscript.

Kind regards,

Muhammad Asaduzzaman, MD MPH MPhil

Academic Editor

Journal Requirements:

2. In the online submission form, you indicated that "The authors declare that they have no competing interests that could be perceived to bias this work.". 

a. In a public repository, 

b. Within the manuscript itself, or 

c. Uploaded as supplementary information.

Additional Editor Comments (if provided):

Reviewers' comments:

Reviewer's Responses to Questions

**Comments to the Author**

1. Does this manuscript meet PLOS Global Public Health’s publication criteria ? Is the manuscript technically sound, and do the data support the conclusions? The manuscript must describe methodologically and ethically rigorous research with conclusions that are appropriately drawn based on the data presented.

Reviewer #1: Partly

Reviewer #2: Yes

Reviewer #3: Yes

2. Has the statistical analysis been performed appropriately and rigorously?

Reviewer #1: N/A

Reviewer #2: N/A

Reviewer #3: N/A

3. Have the authors made all data underlying the findings in their manuscript fully available (please refer to the Data Availability Statement at the start of the manuscript PDF file)?

Reviewer #1: Yes

Reviewer #2: No

Reviewer #3: Yes

4. Is the manuscript presented in an intelligible fashion and written in standard English?

Reviewer #1: No

Reviewer #2: Yes

Reviewer #3: Yes

5. Review Comments to the Author

Reviewer #1: Antimicrobials as cornerstones and quick fixes in healthcare and society: health practitioners´ critical reflections on two stories of antimicrobial use as part of antimicrobial resistance (AMR) education

The manuscript is interested in exploring health practitioners' understandings of the role of antimicrobials in healthcare and society and how this could inform antimicrobial resistance (AMR) education. Applying participatory research workshops, the authors interest is centered in discussing two conceptual stories of antimicrobials in healthcare and society, as cornerstones which are key to the sustainability of healthcare and viability of modern medicine, and quick fixes that are used to mitigate but not resolve deeper and structural challenges as part of healthcare systems and society. Three interconnected themes were identified as part of the results outlining how participants operationalised the two stories as part of AMR education. These included (i) preventing common infections, (ii) addressing risk factors, and (iii) engaging with societal inequalities.

The manuscript presents a different perspective on a relevant topic; however, it is considered important to provide a theoretical and methodological framework. Major and minor revisions are described below.

Major conceptual aspects:

It is important that authors define the object of their study and its scope when making arguments, theoretical discussions and conclusions. Because it is a qualitative study, it is advisable to contextualize/limit the framework of their research. For example, the main global causes of morbidity and mortality are non-communicable diseases, while in developing countries, communicable diseases, closely linked to the use of AMR, are predominant.

Considering the international audience of the journal, it is important to describe the social and health dynamics of the study area, specifically in the functioning and structure of the health system. For example, are there primary health care policies? Are there levels of health care? Why is the study carried out in the selected hospitals? These are some of the concerns that arise when trying to understand the social and health framework that runs through the use of AMR and education.

Finally, a thoughtful debate on the solid focus on the social determinants of health and the “risk factors” mentioned in the manuscript would enrich the theoretical discussion.

Major methodological aspects:

It is desirable that the authors specify the methodological approach of the manuscript. From the reading, it is interpreted that this is a qualitative study. The objective allows us to assume that it is in an exploratory phase of the analysis, however, the scope of the same (healthcare and society) and the conclusions of the authors exceed this methodological framework. This suggestion is valid for the manuscript in general, for example, in the case of the title it is desirable that the authors make reference to the place where the study is carried out. In addition, the objectives should mention the place and time frame in which the research takes place.

It is necessary to specify the arguments that justify the study in the defined area and the selected hospitals (what population do they serve? What is their size? Is there a criterion based on health demand? Are these health centers with the highest record of AMR prescription?).

Qualitative studies are interested in obtaining a deep understanding of people's perspectives in relation to phenomena that occur in a specific place and time. This gives them a different scope from quantitative studies. In this sense, it is contradictory to refer to "health systems and society."

It is desirable to briefly describe the method of instrument validation and specify how the discourse analysis was carried out.

Minor revisions: Please review the general writing of the manuscript to make the text more fluid and avoid redundancies. Consider the length suggested by the journal for the different sections.

Reviewer #2: The introduction lacks clarity about Antimicrobial Resistance (AMR), and a proper scientific definition is missing.

The methodology section requires a detailed explanation of the Participatory Rural Appraisal (PRA) methods and their relevance to this study.

The demographic profile of the study participants must be included.

The study overlooks the impact of cultural and social taboos on health decision-making.

The manuscript needs a more extensive literature review to enhance its scientific depth.

Numerous grammatical and typographical errors, particularly in capitalization, need thorough correction.

Reviewer #3: This is a technically sound, clear and well-prepared manuscript. The findings are mostly clear and supported by clear figures. The data availability statement indicates that data are not publicly available to ensure confidentiality but could be made available on reasonable request. I have a few comments which I hope will improve the manuscript, some minor comments related to grammar, while others require some more substantial revisions (e.g., details on qualitative data analysis and restructuring of quotes/results):

Abstract

* line 23 - ‘from’ two Zimbabwean hospitals, not ‘for’

Introduction

* line 84 – ‘switch it’ is unclear

Materials and methods

* could you say a bit about why Zimbabwe, and whether this work is part of/builds on ongoing work in the study area? Some justification for the particular sites would be useful and a brief outline of the ongoing work (line 111/112 refer to previous research, it would be useful to briefly outline these studies and their findings).

* please add a section on how data analysis was undertaken (of transcripts, field notes and written documentation), linking with the participatory method identified – this is important for understanding the results.

* line 143-145 sentence appears incomplete, please revise.

* would you be able to include details of ‘the two stories’ – how were they developed and how were they used in the workshops (i.e., how were discussions initiated around the images in Figure 1)?

Results

* lines 203-205 – would suggest choosing one quote from the selection that best evidences your point, and include additional details from the quote in the narrative prior to the quote if they are important to include.

* same as above re. the multiple quotes used in the results.

Discussion

* could you discuss how you envisage AMR education being enacted, i.e., is this through CPD or curricula training, or community awareness events. Who is delivering/receiving the education? Are there any barriers to delivery of the education proposed?

* it would be useful to situate the findings in the context of other education initiatives/models for AMR education in the country/region.

* line 406, delete ‘Collingon’

* line 413, should read Ahmed et al. and Coque et al.

* line 415, repetition of ‘for’

Conclusion

* line 430-434, long sentence and difficult to follow, consider breaking this up.

Acknowledgements

* line 458, missing closed bracket

6. PLOS authors have the option to publish the peer review history of their article (what does this mean? ). If published, this will include your full peer review and any attached files.

**Do you want your identity to be public for this peer review?** For information about this choice, including consent withdrawal, please see our Privacy Policy .

Reviewer #1: No

Reviewer #2: No

Reviewer #3: No

---

## [Decision Letter · Decision Letter 1]

Antimicrobials as cornerstones and quick fixes in Zimbabwean healthcare and society: health practitioners´ critical reflections on two stories of antimicrobial use as part of antimicrobial resistance (AMR) education

PGPH-D-24-02902R1

Dear Martin Mickelsson,

We are pleased to inform you that your manuscript 'Antimicrobials as cornerstones and quick fixes in Zimbabwean healthcare and society: health practitioners´ critical reflections on two stories of antimicrobial use as part of antimicrobial resistance (AMR) education' has been provisionally accepted for publication in PLOS Global Public Health.

Best regards,

Muhammad Asaduzzaman, MD MPH MPhil

Academic Editor

Reviewer Comments (if any, and for reference):

Reviewer's Responses to Questions

**Comments to the Author**

1. If the authors have adequately addressed your comments raised in a previous round of review and you feel that this manuscript is now acceptable for publication, you may indicate that here to bypass the “Comments to the Author” section, enter your conflict of interest statement in the “Confidential to Editor” section, and submit your "Accept" recommendation.

Reviewer #2: All comments have been addressed

Reviewer #3: All comments have been addressed

2. Does this manuscript meet PLOS Global Public Health’s publication criteria ? Is the manuscript technically sound, and do the data support the conclusions? The manuscript must describe methodologically and ethically rigorous research with conclusions that are appropriately drawn based on the data presented.

Reviewer #2: Yes

Reviewer #3: (No Response)

3. Has the statistical analysis been performed appropriately and rigorously?

Reviewer #2: N/A

Reviewer #3: (No Response)

4. Have the authors made all data underlying the findings in their manuscript fully available (please refer to the Data Availability Statement at the start of the manuscript PDF file)?

Reviewer #2: Yes

Reviewer #3: (No Response)

5. Is the manuscript presented in an intelligible fashion and written in standard English?

Reviewer #2: Yes

Reviewer #3: (No Response)

6. Review Comments to the Author

Reviewer #2: All comments and recommendations have been incorporated by the authors thoroughly

Reviewer #3: No further comments, thank you for addressing the reviewers' comments in such depth

7. PLOS authors have the option to publish the peer review history of their article (what does this mean? ). If published, this will include your full peer review and any attached files.

**Do you want your identity to be public for this peer review?** For information about this choice, including consent withdrawal, please see our Privacy Policy .

Reviewer #2: No

Reviewer #3: No
